# The Museum as a Laboratory: An Approach to the Experience of Public Museums in Chile

**Marisol Facuse Muñoz** [1,*] **and Raíza Ribeiro Cavalcanti** [2,*]

1 Department of Sociology, Sociology of Art Group, Faculty of Social Sciences, University of Chile, Santiago 7760210, Chile
2 Department of Publicity and Image, Technological Faculty, University of Santiago, Santiago 9170022, Chile
* Correspondence: marisolfacuse@uchile.cl (M.F.M.); raiza.ribeiro@usach.cl (R.R.C.)

**Abstract:** The present article analyzes the recent debates regarding the redefinition of the museum, exploring resonances in reflective practices and processes in public museums in Chile. While these have caused controversy and discord, they appear to converge in the need to rethink the relationship between museums and society, seeking to make them more inclusive, democratic and diverse. The present discussion is based on the preliminary results of "LAB_Museums: Contemporary Museums and Museologies", an ongoing interdisciplinary research intervention model promoting processes of co-production of knowledge regarding museums and museography. This paper is the publication of the results of the project. To this end, a collaborative ecosystem of knowledge has been developed between the university, museums and public sector, based on the implementation of laboratories, initially, in five public museums of the Metropolitan Region of Santiago, Chile: the National Museum of Fine Arts, the National Historical Museum, the Museum of Contemporary Art, the Popular Art Museum and the National Center of Contemporary Art. The theoretical/methodological framework used was that of Institutional Analysis (IA), based on which interviews and discussion groups with museum professionals promote dialogues on the present reality and contemporary challenges of museums.

**Keywords:** public museums; participatory museums; contemporary museology; institutional analysis





## 1. Introduction

Over the last decade, the museum has been the object of an intense process of redefinition in the theoretical space as well as the institutional one (ICOM 2019a, 2019b; 2022a, 2022b; Rasse 2017; Brulon 2015; Silverman 2014). In this way, concepts such as the participative museum (Message 2014); the community museum (Rivière 1989; Crooke 2008; De Varine 2016, 2017); the reimagined museum (Witcomb 2003; Rasse 2017) and the hybrid, feminist or decolonized museum (Rechena 2014; Cortés and Bermejo 2019) have opened up new ways of conceiving museums and their relationships to society.

Considering the multiple dimensions and levels of this debate—theoretical, institutional, global, local and community—the present article will explore the resonances of this discussion in the concrete practices of the public museums of Chile. With this in mind, we seek to answer the broader question of how the debates on the redefinition of museums are influencing or contributing to the new local museum dynamics. Is there continuity between these debates and the concrete action of museums in Chile or, rather, is it a debate encapsulated in the elites of the Global North?

Starting from the notion of the museum as a dynamic and self-reflective space, we can observe the continuity and ruptures between these debates and the daily reality of Chilean museums, taking as a case study the principal public museums of the country. Based on the perspective of Institutional Analysis (AI), we can analyze how museum institutions participate in these debates on identity, social functions and the future of museums concerning society's challenges. The preliminary results that we present in this

paper are in the framework of the first phase of an ongoing project named "LAB_Museums: Contemporary Museums and Museologies", an interdisciplinary research laboratory based on a collaborative methodology between teams of academics, public sector and museum professionals. The methodological perspective employed in this stage was a study of multiple cases consisting of five museum institutions: the Museum of Fine Arts, the National History Museum, the Museum of Contemporary Art, the Tomás Lago Popular Art Museum and the National Centre of Contemporary Art. (Figure 1).

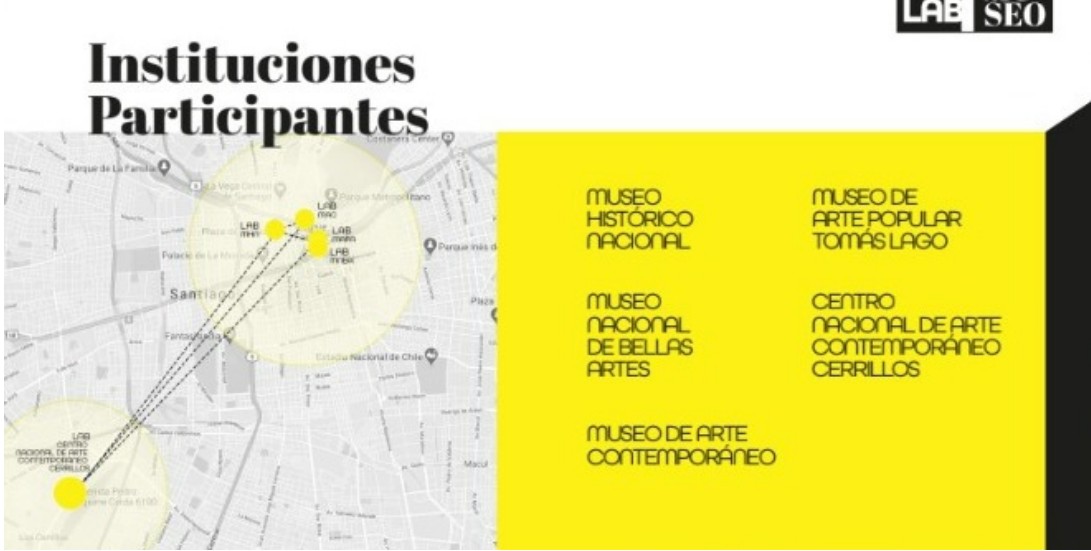

**Figure 1.** LAB_Museums case study. Designer: Yasmin Fabris (UFPR).

The discussion that we propose based on this case study aims to contribute to the field of study regarding museums from the perspective of Institutional Analysis (AI), restoring the importance of the communities of which the museums are a part for the processes of reflection on museum dynamics in the present. With that, the centrality of the role of professionals of museum institutions as central agents becomes central to the institutional transformation processes (Kann-Rasmussen and Hvenegaard Rasmussen 2020), observing aspects such as the organizational autonomy and specificity of the museum and its transformation in the perception of the actors themselves. To this end, we will analyze the encounters and discussion groups held in the framework of the first stage of the LAB_Museums project to clarify how the transformations are being negotiated and elaborated in the museological field on a global and local level by the professionals of the museums who vitalize, modulate, reproduce and/or modify the institutions in their daily duties.

## 2. The LAB_Museums Project: Collective Thinking on Museologies in Chile

The LAB_Museums project began in 2020 with an internal call by the University of Chile for the conformation of the Laboratories in Humanities (UCH-1899). The conception of the project coincided with two crucial moments: the first was the process of the global redefinition of museums (ICOM 2019a, 2019b) and the second was the declaration of the global health emergency due to the COVID-19 pandemic. The latter limited the implementation of face-to-face activities in museums and territories, but conversely was a factor that promoted greater internationalization of both universities and the institutions of museums.

In its initial phase, the project oriented its action towards five public museums of the Santiago Metropolitan Region, beginning with those that promoted active dialogue with researchers from universities of different continents, as well as professionals from museum institutions through the model of collaborative research (Desgangé 2007; Morrissette 2013).

The methodology was based on various types and occasions of dialogue, including diagnostic meetings on the current situation of participating museums, workshops for museum workers, talks for the general public and the production of digital content.

The five museums participating in this first sample share the common characteristic of being national museums (the National Museum of Fine Arts, the National History Museum), university museums (Museum of Contemporary Art, Tomás Lago Museum of Popular Art), in addition to a center of contemporary art (National Center of Contemporary Art), all of which are in the Metropolitan Region and are well established in the national museum sector. The desire to initially incorporate these museums was to observe how both local social conflicts (the popular uprisings of October 2019 in Chile) and the changes in museological paradigms on a global level reverberate in these institutions (national, nineteenth century, etc.).

At the end of this first phase, we advanced to the second stage, in which we replicated the Lab_Museums design on a regional scale based on the collaboration between networks of museums of the Ñuble and Valdivia regions, as well as universities and local cultural institutions. This phase is currently in the process of execution, in which we are establishing networks and processes of collaboration in the development of activities such as workshops and seminars, in addition to diagnostic meetings with workers from the previously mentioned museum networks.

The data elaborated on in the present paper refer to the diagnostic meetings (focus groups) (Figure 2) held during the first phase of the realization of LAB_Museums, which coincided with a period of profound socio-political changes in the country: social upheaval, a constitutional process, the pandemic, etc. The reflections on the part of the museum professionals during that time provide important data for the analysis of how some of the practices of these professionals brought about the involvement of the museums in the political process of the moment, despite the conservative structure of their institutions.

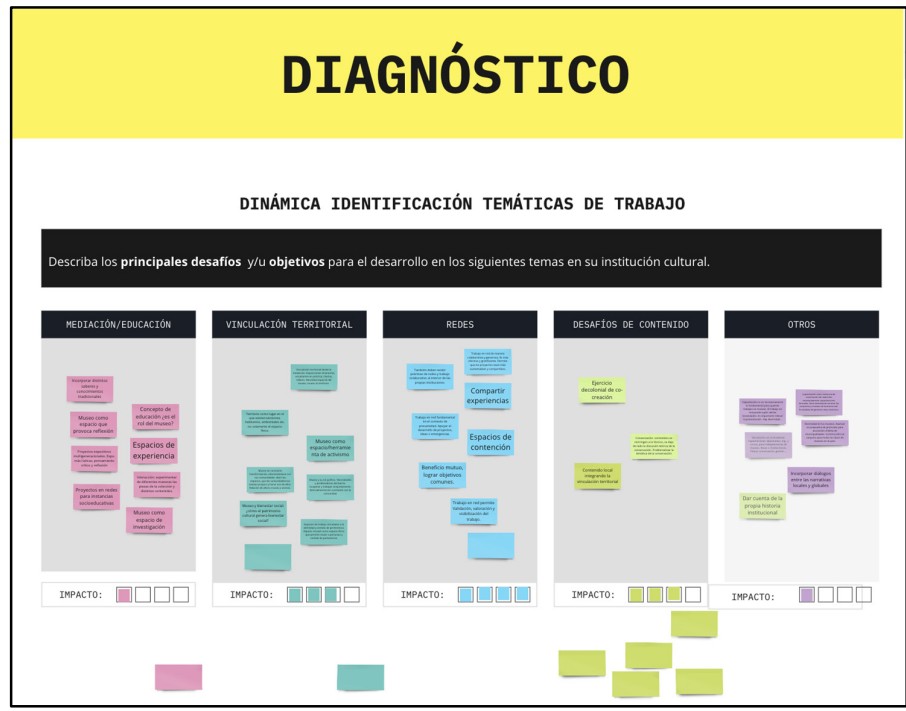

**Figure 2.** Table of contents identified in diagnostic meetings. Designer: Yasmin Fabris (UFPR).

## 3. What Is a Museum? Current Controversies Regarding a Definition

On a global level, one of the most important milestones of the debate on the identity of the museum took place at the International Council of Museums General Assembly (ICOM 2019a), which gathered the principal museum institutions to agree upon a new

definition that centered on the role of museums in contemporary societies. Beginning with a participative process that began in 2016, various museums around the world proposed concepts and ideas for the new definition. In this context, the need forcefully arose to rethink museum practices so that these institutions would become more relevant for communities, considering their responsibility "to establish dialogue between cultures, to build bridges for a peaceful world and to define a sustainable future" (ICOM 2019a), and understanding them as polyphonic spaces, community articulators and heterogenous memories.

However, far from generating a consensus, the occasion presented significant theoretical controversies and policies when answering the question "What is a museum?". Some representatives argued in favor of a profound transformation of museums, emphasizing the need for a definition radically different from the existing one, making the museum a more democratic and inclusive space:

> Museums are democratizing, inclusive and polyphonic spaces for critical dialogue about the pasts and the futures. Acknowledging and addressing the conflicts and challenges of the present, they hold artifacts and specimens in trust for society, safeguard diverse memories for future generations and guarantee equal rights and equal access to heritage for all people. Museums are not for profit. They are participatory and transparent, and work in active partnership with and for diverse communities to collect, preserve, research, interpret, exhibit, and enhance understandings of the world, aiming to contribute to human dignity and social justice, global equality and planetary wellbeing. (ICOM 2019b)

The discussion of a new definition of the museum emerged from a deeply critical view of what had hitherto prevailed, as it was believed to be anchored to past values, and was requested to be "historicized, contextualized, denaturalized and decolonized" (Sandhal, in: ICOM 2019b). The proposed definition of 2019 (ICOM 2019a, 2019b) meant a radically different perspective, projecting a new direction for museums, prioritizing equality of rights and access to heritage for all people (Kioto, ICOM 2019b). This was supported by the Scandinavian countries, Australia and the United Kingdom. However, the use of progressive terms such as "planetary wellbeing" and "social justice" caused friction among the national representatives of countries such as France, Italy, Spain, Germany, Canada and Russia, who saw in the definition an "ideological manifesto" (Schope 2020, Jasmine Liu 25 August 2022).

On the contrary, other positions were less inclined to deviate too much from the prevailing definition, holding to the belief in the need for the preservation of the specificity of the museum as a cultural institution that maintained the functions of research, collection, preservation and exhibition of heritage, and not those of other institutions such as libraries, cultural centers or even NGOs, thus adding a geopolitical dimension to the debate.

The inability to reach an agreement resulted in the suspension of the debate until 2022, when, at the ICOM Convention in Prague, a new definition was agreed upon which, although integrating novel elements, did not represent much of a deviation from that which has prevailed for the preceding decades:

> "A museum is a not-for-profit, permanent institution in the service of society that researches, collects, conserves, interprets and exhibits tangible and intangible heritage. Open to the public, accessible and inclusive, museums foster diversity and sustainability. They operate and communicate ethically, professionally and with the participation of communities, offering varied experiences for education, enjoyment, reflection and knowledge sharing." (ICOM, Prague, on 24 August 2022)

The Extraordinary General Assembly of the ICOM has approved the proposal for the new museum definition with 92.41% (for: 487, against: 23, abstention: 17).

Nevertheless, the wide-ranging support obtained does not resolve the coexistence of opposing positions at the heart of the global institutions dedicated to the sector. These controversies uncover the controversial nature of the museum as an institution, revealing

the various concepts and ideologies revolving around controversial themes such as substantive gender equality in museums, the relationships between the Global North/South and the processes of decolonization and restitution of pieces, bodies and artifacts. (Arthur and Ayala 2020).

For its part, the 2019 definition was enthusiastically received in Latin America. In the case of the ICOM of Chile in the same year, it brought together a group of professionals and researchers from different regions to debate a new definition for museums. The methodology consisted of comparing different definitions of museum, including the ICOM's 2019 proposal and the Museo Integral, emanating from the 1972 Santiago Round Table (Nascimento et al. 2012a, 2012b), which saw the museum as an institution committed to the processes of social transformation: "an institution at the service of society, of which it is an inalienable part, and that is in its very essence the elements which allow it to participate in the formation of awareness among the communities (...) committing itself to the prevailing structural changes and causing others within the respective national reality" (ICOM-Chile 2021).

The local discussion coincided with the period of the social uprising of October 2019, characterized by the demand for more social rights and a life of dignity. In accordance with this sentiment, the discussion highlighted the need for greater social participation in museums, demonstrating historical continuity with what was put forward during the Santiago Round Table of 1972 (UNESCO 2021, 2022) and incorporating new themes of current social revindication such as feminism, sustainability and indigenous rights, proposing a new definition of concepts related to the museum, such as communities, collective memory, participation, gender equity and decolonization, among others.

## 4. The Museum through the Prism of Institutional Analysis (IA)

The theoretical focus through which we viewed the museums that we studied in this article is that of Institutional Analysis (Guattari 2003; Lourau 2004; Hess 2004; Lapassade 1974; Castoriadis 1975), with which we seek to comprehend the nature and singularity of the institution of the museum as a Western invention that has been acquiring a global dimension over the last three centuries. It is important to mention that the perspective of Institutional Analysis (IA) to which we adhere is not entirely based on institutional and neo-institutional studies of politics and economics, the central focus of which has been the analysis of individual behavior in contrast with the institutional environment based on assumptions such as that of the rational actor. Our analysis is closer to that of the perspective derived from institutional studies in the context of hospitals and educational establishments, inviting us to conceive of institutions as processes of permanent creation and self-reflection.

Based on Institutional Analysis, the institutions are understood as realities undergoing constant transformation that articulate the dimensions of the instituted and the instituting (Lourau 2004). This means rethinking the boundaries of an institution (the city, the school, the trade union, the hospital and the museum), comprehending it in its inherent duality: as a space of projection (the instituting) and as a collection of norms and counter-norms that can always be renegotiated (the instituted). To analyze the institution means to observe beyond its constitution (its infrastructure, its norms and its legal aspects), understanding it as a collection of practices and processes of reflection on these practices and considering its subjective dimensions and the desires that are projected from these practices (Schaepelynck 2018). In other words, it is not always the rigidity and impermeability commonly associated with the institutions that define daily reality. Thus, museums, like any other social institution, are constituted of conflicting discourses, practices, materialities and ideologies that make political spaces of them which are transited by multiple agencies. This way, we will analyze the institution of the museum beyond its establishment. In doing so, we will take into account the multiple dimensions of its practices and the reflections of the actors on these practices, understanding the museum as a dynamic institution that can be adapted to the demands of new socio-historical contexts.

Recently, this perspective has become of renewed institutional ethnographic interest, proposing a situated and relational view from its subjects in various institutional contexts (Jirón et al. 2018; Klostermann 2020). With this focus, this study intends to contribute to a phenomenological perspective in order to comprehend the processes of change in the institutional practices of museums, which have so far been mostly centered on focuses that prioritize the analysis of cultural policies, management, the analysis of collections, frequentation, the characterization of the public and cultural participation (Darbel and Bourdieu 2007; Bourdieu 2010; Antoine 2012; Antoine and Carmona 2014, 2015).

Of great importance to our analysis will be the contributions of neo-institutionalism, especially the approach of Powell and Di Maggio (1999), who distinguish between the institution and the organizations. From this point of view, the institutions are like the rules of the game, i.e., the norms and values on which the organizations are based, which make up the structured whole of individuals in action (Powell and Di Maggio 1999). The aforementioned is important considering that it is of interest to us to observe the museum in its organizational bases, from the practices of its professionals and their different roles in order to comprehend how the processes of change and tension experienced by this institution interfere with their particular organizational cultures, redefining their *Values Framework* (Davies et al. 2013).

## 5. Collaborative Research Methodology

As we have seen, the main objective of the LAB_Museums project has been to generate processes of institutional and scientific reflection on the current reality of the museums in which the researchers and professionals participate in the implementation of a model of collaborative research (Desgangé 2007; Morrissette 2013). Based on the study of multiple cases (Yin 2018), a sample of five public museums of the Metropolitan Region of Santiago was taken, including the National Museum of Fine Arts, the Museum of Contemporary Art and the National Centre of Contemporary Art. Based on the interviews and discussion groups with authorities and professionals, it sought to analyze how the debates on the definition and future of the aforementioned museums can cause transformations in the dynamics of local museums.

The collaborative research model turned out to be the most suitable for orienting our research given its proximity to the Institutional Analysis perspective (Lapassade 1974), which makes the focus of the agents relevant. According to Morrissette (2013), "Collaborative research is a methodology of intervention which supposes that professionals are committed, along with the researcher, to exploring an aspect of their practice and that the very objective of the research has repercussions in the contextual comprehension of this aspect" (Morrissette 2013, p. 25). In our research, this focus offered new ways to initiate a process of co-production between social scientists and actors of the heritage institutions. From this perspective, we brought about an immersion in the practical world of museum professionals, promoting processes of reflection on museum practices and on the "know-how" related to these practices. The implementation of this collaborative research consisted of three cycles beginning with the **diagnostic cycle**, followed by the **cycle of cooperation**, and finally, that of **co-production**.

During the **diagnostic phase**, discussion spaces were generated with focus groups which brought together museum professionals with the aim of identifying the most important current problems and challenges. Afterward, in the **cooperation cycle**, seminars and meetings were held to discuss the main problems and challenges identified during the previous cycle, presenting international experiences and resolution strategies with regard to the problems presented during the previous cycle from a comparative perspective. A central element to this second cycle was the configuration of an international network made up of universities and museums in Chile and the rest of the world, which was increasingly complicated not only in its levels and scales of action but also in its geographical coverage, as shown in the Figure 3.

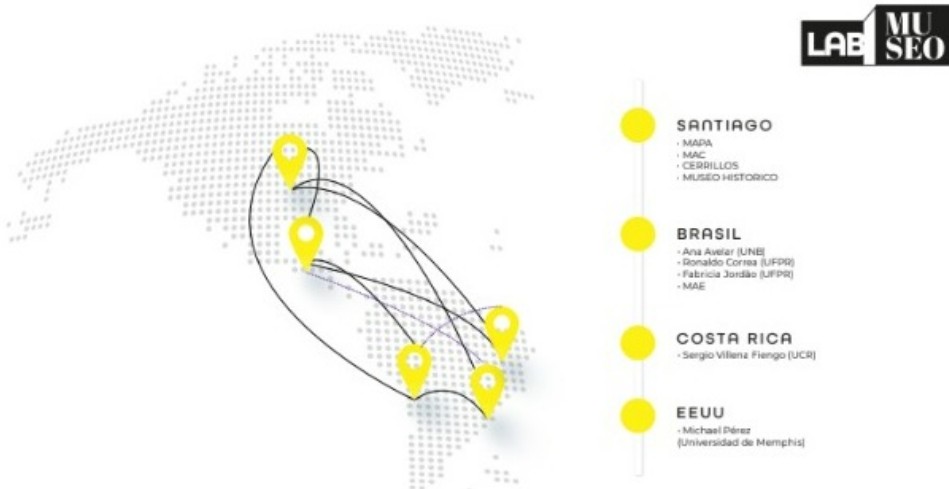

**Figure 3.** LAB_Museums international networks. Designer: Yasmin Fabris (UFPR).

This network was the basis of the "institutional encounters" (Manero 2018) in which open dialogues were promoted with minimal direction, thus generating a "collective analysis" in order to invite all of the participants to assume the role of "psychoanalysts" and "analyzers" of the institutional practices, collectively generating interpretations and responses regarding the present and future of museums (as can be seen in Figure 4). For Lapassade, this is a process of "reconciliation between the analyzer and psychoanalyst that can only be realized at the moment in which all become psychoanalysts and analyzers at the same time" (Manero 2018, p. 139).

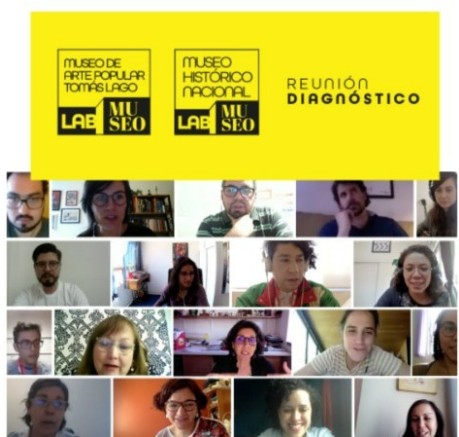 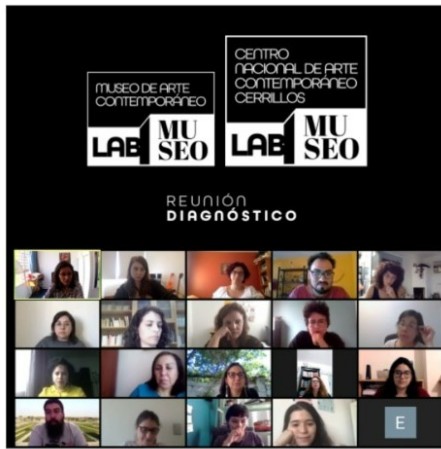

**Figure 4.** LAB_Museums diagnostic meetings. Designer: Yasmin Fabris (UFPR).

The discussions arose from the various provocative questions that not only inspired dialogue but at the same time offered breadth and liberty in the selection of themes of interest. In other words, an exercise of "institutional therapy" (Manero 2018) in order to identify collective responses to the questions "What is the social role of museums?" and "How do we want museums to exist in the future?"

The analysis of this initial corpus of the discussion allowed us to provisionally organize the topics into the three most frequently recurring general categories in the discussion: (1) Institutional challenges; (2) museums and social crisis and (3) museums and communities. In the following section, we will present the initial results of the analysis of the content of the discussions held during the three phases of the study.

*5.1. Institutional Challenges*

The institutional aspect of the museum emerged in various interventions and reflections during the LAB_Museums focus groups. In part, we can attribute this phenomenon to the discussion on the definition of museums that continues to cause controversy on an international level. However, on the other hand, the discussions indicated various aspects that were more local and particular to the institutions of museums and of Chile in relation to the social and political contingencies of the country.

In encouraging the workers to assume the position of "analyzer" (Manero 2018), an interesting reflection was made that discussed institutional roles, the definition (or lack of definition) of functions, the relationships between different educational themes, inter-institutional relationships and the need for new institutional forms or models of museums.

Analysis by workers emphasized the fundamental importance of a greater organizational structure without which duties remain undefined and even rendered impossible. One reflection makes visible the need to reconcile the current debates regarding the definition of museums with the organizational and functional aspects of the institutions, which ultimately affect their workers in their practices and objectives. This aspect reveals issues for the analyst/analyzer of institutions regarding the articulation of debates on the redefinition of the museum with the organizational structures that currently constitute its materiality.

The comments on organizational conflict and the division of tasks within the institution illuminate the changes within the Museum Values Framework (Davies et al. 2013). The detailed disputes reveal, in the practices of the professionals and their internal conflicts, how these values are promoted in the institution and how they bring about change. This is perceived in the discourse which demonstrates that, while some of the professionals tend to look for changes in the exhibition outlines and themes, others seek to conserve a more conservative image or structure within the institutions, causing tension and disputes.

Another aspect is the institutions' limited budgets, which make it difficult to project institutional actions for a greater period of time. The difficulty in projecting activities beyond a certain period of time coupled with the budgetary constraints include, as a consequence, the aspect of job insecurity and ends up becoming an enormous obstacle to the democratization of the institutions of museums. All of the necessary organizational, occupational and intellectual processes to launch activities such as the broadening and diversification of the public, the dissemination of the content of expositions and the territorial and community connections, among others, are severely affected by these financial limitations to the point of, in some cases, impossibility.

Difficulties in financing have affected all of the institutions to a greater or lesser extent, impacting different areas from programming to the educational sector and collections. In contrast with countries such as the United States, where there is a more established tradition of private patronage of cultural institutions, the tendency is still more tilted towards their public financing in Chile. However, the process of neoliberalization of the country has made the financing of culture dispensable, generating situations of *impasse* in which the employees of the institutions are made to look for alternative forms of patronage in order to carry out their tasks effectively or to outsource tasks that are central to the functioning of the institution.

On the other hand, there was an equally notable difficulty in recognizing the work of educational sectors in institutions, a factor that directly affects the relationship between museums and the public. While it is certain that the current importance given to the public is an inevitable shift for museology (be it from the perspective of cultural democratization or from a more managerial focus of museums), this concern does not always translate into greater importance given to units of mediation or education towards this end. As a result, mediation efforts are, for the most part, most affected by job insecurity and are concentrated mostly on the voluntary services of museums, as noted by the works of Susan Ashley (2016), Cintia Silva (2017) and Carolin Sudkamp (2021).

Finally, it is important to highlight the political and social period marked by contingencies such as the social uprising of October 2019 in Chile and the COVID-19 pandemic,

which deepened certain processes of institutional self-reflection that had been forming, especially in national museums. The birth of these institutions along with the very nation calls their narratives into question in light of the debates on decoloniality. This is the case of the National Museum of History, which is the impetus behind the process of changing the outline with a tendency towards the incorporation of a more plural concept of nation.

More specifically, the aspects highlighted by each institution on this issue can be summarized in the following manner, as we can see in Table 1:

**Table 1.** Institutional Challenges.

| Institution | Central Issue |
| --- | --- |
| National Museum of Fine Arts | Undefined organizational chart, overlapping functions and internal dialogue difficulties. Processes of reviewing collections, mission and museum policies (intensified by the social upheaval and the COVID-19 pandemic) |
| National History Museum | Deep institutional reflection. Change in the script of the permanent exhibition impacting the mission and policies of the institution. |
| Museum of Contemporary Art | Economic precarity hinders the development of a more systematic programming and the implementation of actions. |
| Tomás Lago American Museum of Popular Art | Difficulty in inventorying the museum collection: limited staff and a tight budget. |
| National Center for Contemporary Art | Lack of adequate permanent staff, constant need to rely on external professionals to fulfill core tasks. |

*5.2. Museum and Social Crisis*

As we have seen, the paradigm of new museology has generated new conceptualizations and reflections on the social and symbolic functions of the museum. These debates have gradually brought museums closer to the revindication of new social and political actors. Accordingly, feminists, Indigenous people, sexual dissidents, and immigrant communities have been entering the museum through the promotion of new artistic scenes, the reimagination of outlines and collections or the generation of actions and spaces of dialogue that restore the function of the museum as that of a public hall. This kind of action has caused many actors to speak of "activist museums" (Message 2014) to account for the implication of cultural institutions in social movements and the struggle for recognition.

The activist museum is, for Kylie Message, that which recognizes the role of culture in the processes of socio-political transformation and the struggles for rights; especially in the case of the United States, the struggle of indigenous people for recognition. In her words:

> "My main focus is on the ways in which the national history museum responded to demonstrations for social, political and economic reform (such as the 1968 poor people's campaign). This means I have analyzed how it has been framed by, and how it has responded to, external debates about social, political and economic reform, including constitutional issues and citizenship." (Message 2014, p. 3)

A more meticulous observation of the cases studied shows the effects of the 2019 social uprising in Chile on the cultural institutions, with museums called upon by citizens to assume a more active and democratic role. In this context, various institutions assumed a leading role in the debate on greater equality and social rights. Particularly noteworthy in this context were the activities organized by the National Museum of History and the Museum of Fine Arts, which proposed a community-based and democratic review of its day-to-day operations in the context of the social uprising (Facuse and Cavalcanti 2020).

One example is the notion of the *museum forum*, put forward at the time by a worker at the National History Museum, which resonates with the idea of the *museum platform* mentioned by the team of the institution during the focus group and shows how the institution is projected: as a space open to dialogue and debate, where discourse is not

reproduced vertically from top down but circulates from the museum to its public and vice versa. This exercise of institutional dialogue mentioned by the worker is materialized in actions such, as we can see in Figure 5, for instance, the invitation of local neighborhood residents to participate in town hall-type meetings, i.e., citizen meetings that took place all around the country during the social uprising in order to think collectively about the constitutional process.

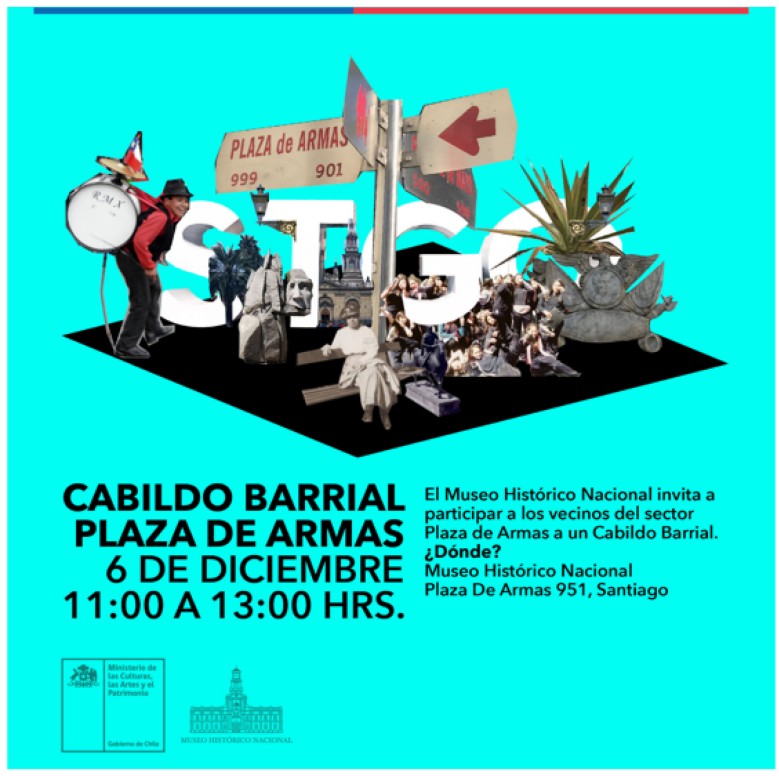

**Figure 5.** Neighborhood assembly invitation (2019). Source: National Historical Museum.

The workers of the National Museum of Fine Arts likewise assumed positions during the political processes in the country, causing the museum to engage with the contingency and participate in a moment of strong social criticism. An interesting action carried out by the institution during the days of the social uprising was to create a space for public comments on the museum, as the example in Figure 6. Most messages for the museum called for it to think of itself as a space more open to the public. This listening exercise was extended to the gathering of opinions from the public on the graffiti written on the walls of the institution during the protests with a banner raised in the hall of the museum. It is also worth highlighting that in some museums both the directors and the workers of the museums affected by graffiti sought to generate participative processes based on these slogans, opening the discussion to the community and going against the order to remove the graffiti and clean the facades of the museum that came from the Ministry of Culture at the time. Instead, the Museum of Fine Arts installed panels of paper in order to continue receiving intervention from the streets inside its facilities.

With a similar perspective, the Tomás Lago Museum of Popular Art created the "Heritage In Montion" initiative, a convocation to carry out an exposition of elements born of the protest based on contributions from the public. The objective of the call was to generate a collection of visual interventions in various formats: material, visual, graphic, etc. This experience was important for the generation of a mini-process of creating a collection of popular artefacts with a collaborative focus based on the direct contribution of the people and creators themselves. The wide-ranging scope of the call opened up the

possibility of equal participation for artists and non-artists in this collection, which was born of the social uprising.

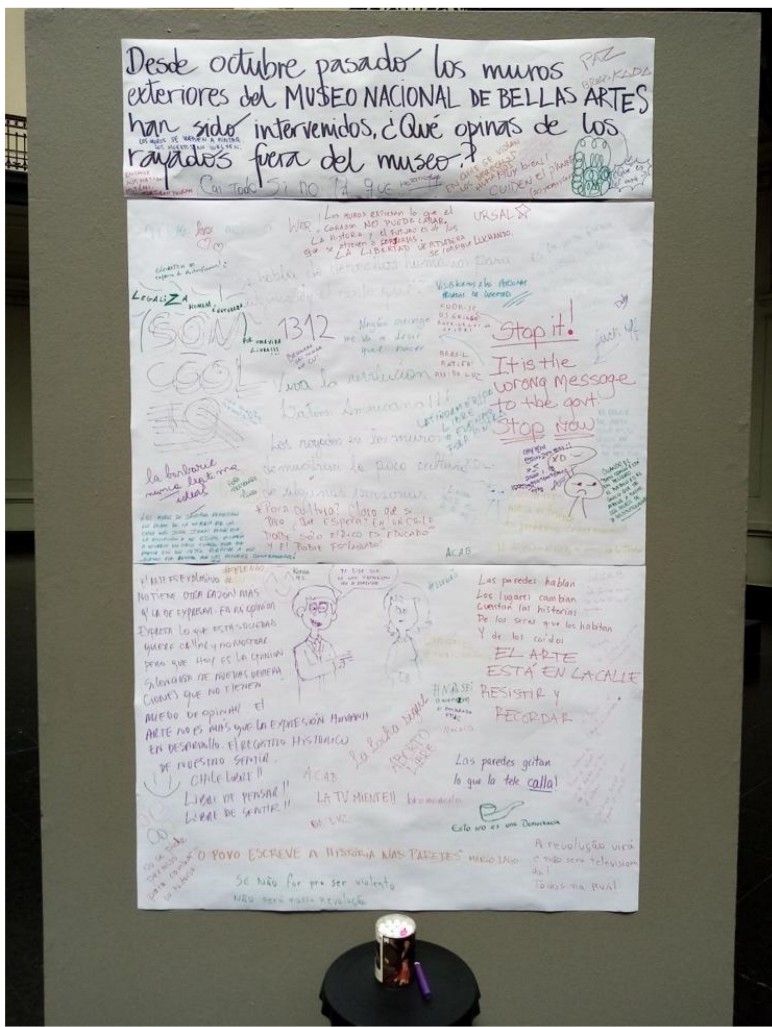

**Figure 6.** Visitor consultation banner on graffiti. Author: Raíza Cavalcanti (2019).

This action is noteworthy for revealing another interesting aspect, which was the debate on the popularity that emanated from this collection that gathered a series of artefacts produced in non-artistic contexts that defy more conventional categorizations of popular art more commonly associated with indigenous people or with popular artists related to traditional cultures, especially in Latin America. This action resulted in an important space for generating mechanisms of participation, calling on the public to participate in an emerging collection that promoted greater institutional legitimacy and participating in a debate on the relationships between popular art and the people, both of which are notions that are undergoing constant transformation, as indicated by this museum's professionals.

Another area of interest was the process of self-organization of the institution's workers as a consequence of the social uprising at the time. One of the initiatives generated at the time was the "Ground Zero Museums", an organic alliance—as defined by a member—that sought to generate a process of mutual support among the workers of the museums, most of whom were located in the main urban points of protest. This group met to collectively consider actions of mediation and dialogue with the public participating in the collective elaboration of the situation of conflict surrounding the institutions, especially the National Museum of Fine Arts, the Museum of Contemporary Art and the Tomás Lago Museum of Popular Art.

Along with this self-organizing movement, the workers of the institutions close to the *ground zero* of the protests equally promoted actions such as performances in front of the institutions, as well as in self-convened strikes, participating in various ways in the social revindication of the uprising, as shown in Figures 7 and 8.

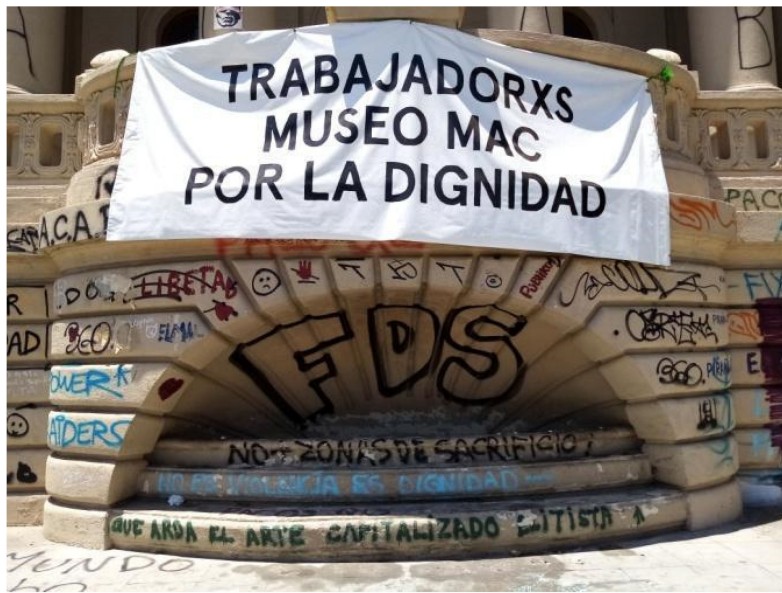

**Figure 7.** Statement by self-organized workers, MAC. Author: Raíza Cavalcanti (2019).

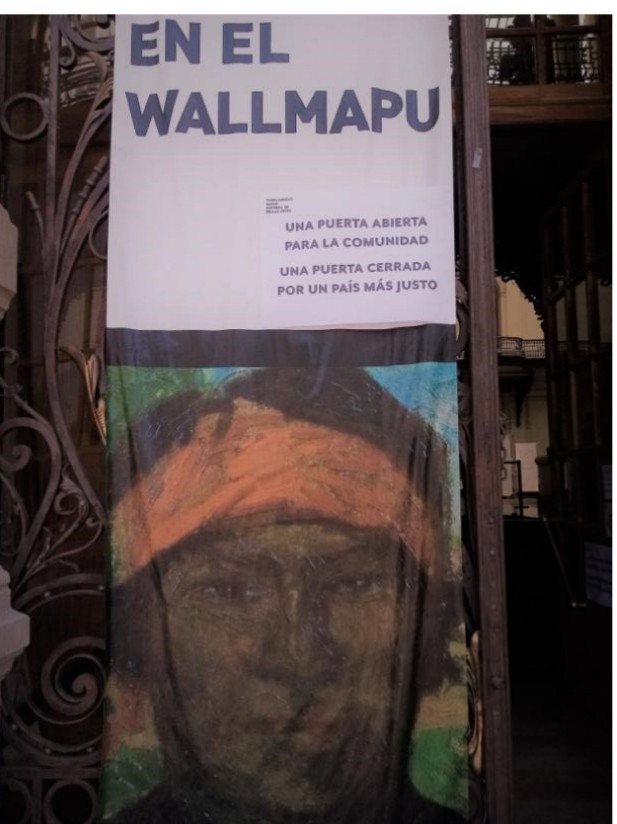

**Figure 8.** Statement by self-organized workers, MNBA. Author: Raíza Cavalcanti (2019).

The aforementioned examples illustrate how the public museums of Chile faced the recent social and health crises with a process of collective self-reflection. Many of the professionals highlighted the importance of participation of the museums in the social

revindication, demonstrating their dialogic, social and political role in seeking a greater connection for these institutions with the dynamics of society.

More specifically, the aspects highlighted by each institution on this issue can be summarized in the following manner, as we can see in the Table 2:

**Table 2.** Museums and Social Crisis.

| Institution | Central Issue |
| --- | --- |
| National Museum of Fine Arts | Implementation of dialogue actions with the public through social media and in-person interactions. Workers promote processes of political reflection in the museum through educational activities, performances and curation of exhibitions with inclusive themes. Organization of workers in the *Museos Zona Cero* (Museums Zone Zero) action and strike by professionals in support of protests. |
| National History Museum | Conducting *Cabildos Ciudadanos* (Citizen Assemblies), engaging neighbors in close proximity to the museum into the constitutional debate. |
| Museum of Contemporary Art | Organization of workers in the 'Museums Zone Zero' action and strike by professionals in support of protests, highlighting internal issues of job precarity. |
| Tomás Lago American Museum of Popular Art | Establishment of the *Patrimonio en Marcha* (Heritage in Motion) call for the inclusion of visual civic expressions in the museum's collection, expanding the notion of 'popular' that it signifies. |

*5.3. Museums and Communities*

A final relevant theme that arose from the dialogues was that of the museum professionals during the diagnostic meetings was the relationship of museums to the communities. Although the very notion of community is complex and must be accompanied by exhaustive comprehension and conceptualization (Crooke 2008), in general, we can understand this term as a group of people consisting of the institution's main prospective user base.

Although for a specific type of museum this definition of community is more easily understood, as is the case mentioned by Kylie Message of the National Museum of the American Indian in the USA, it is more complex and less clear in other cases, as is the case with the National Museums in Chile. Nonetheless, in line with Message, socio-political processes may cause the national museums to be used by certain communities as a means of political representation, even if they are not directly defined by the community-based relationship. In her own words:

> My focus here is on understanding and communicating how and why national museums—specifically those associated with the federal US government and those representing American Indian nations and tribes—have been identified by particular community groups as having utility for their plans to promote community and economic development and recognition of the political sovereignty of Indigenous groups, as well as cultural renewal. (Message 2014, p. 8)

This reflection is important for considering the cases of the institutions of which the LAB_Museums sample group consists, as an imagined community of the Chilean nation is more fundamentally represented mostly by national or university museums. A similarly important reflection was on how a national museum can generate community; before the period of the pandemic, it was unclear to such institutions themselves who the users of the institution even were, i.e., the real people who are part of the community of the museum. It is important to note that the period of the pandemic was important in that the museum could connect to national territories and thus forge links in which the institution becomes relevant to other communities that can be represented and/or see the institution as a means of collective representation, as mentioned by Message.

From the accounts of the LAB_Museums focus groups, it was possible to find conceptual elements that are part of the internal process of institutional reflection carried out by

the National Historical Museum on the occasion of the changing of its script. From this reflection, several interesting studies have emerged, including the especially noteworthy publication "The Mestizo Museum" (Andrade et al. 2018) in which historians Pablo Andrade, Leonardo Mellado, Hugo Rueda and Gabriela Villar reflect on "the national" and "the historical" from the perspective of cultural miscegenation:

> Finally, and in this regard, this idea of multiple time or mestizo time, that is aware of change and continuity but at the same time in coexistence between both, will be recognized and distinguished museographically, disciplinary, and especially museologically, reinforcing with it the idea of "mestizo museology", as well as the relationship of the temporalities with their changes, continuities and its dynamics seen by the historical subjects from their own temporalities. (Andrade et al. 2018, p. 19)

The process of changing the script at the National Historical Museum is a case of great interest for analyzing how national institutions can reconsider their foundations and how this impacts the institution. The review of the museum's permanent exhibition has led to various engagements with groups of people belonging to indigenous communities, as well as with migrant groups residing in Chile, to name a few. The museum aims to promote a listening process to create a new, more inclusive script capable of incorporating narratives from indigenous peoples and people from other territories into the history of Chile.

Furthermore, the elitist aspect of the National Museum of Fine Arts was mentioned, leading to reflection on the active role of the institution in opening up to other communities that do not feel part of the museum. This question refers directly to territory, which is directly related to the notion of community. From the National Museum of Fine Arts, a profound reflection was observed on self-recognition as an institution pertinent to a territory marked by inequality and processes of social precariousness. This recognition affected, according to professional accounts, how the museum is rethinking its history and collections and reviewing its practices (protocols, ways of operating) in order to reconsider how it presents itself to the community. That this reflection seeks to place the museum in a "real" territoriality and not just an imagined one—that of Chile as a nation—is of great importance in determining who it is designed for and thus generating more inclusive and democratic dynamics.

From the Tomás Lago Museum of Popular Art, it was noteworthy how the museum recognizes a community more directly involved with the museum consisting of popular craftspeople and artists, with whom it seeks to constantly maintain contact through talks, workshops and open invitations for craftspeople such as the "seal of excellence in craftsmanship", from which they can publicize their works and even have them exhibited and incorporated into the collections of the museum. However, with the social contingency, a reflection on how the museum could become more socially relevant was made, extending this community to the citizens and producers of visual artefacts made during the protests in consideration of the fact that this museum receives an ample public ranging from school children to tourists and visitors to Barrio Lastarria, a tourist area undergoing strong gentrification. The reflection generated brought about the rethinking of the very notion of the popular, giving meaning to the collections and seeking to open itself up to other types of production.

In the case of contemporary art museums, the relationship with the community is proposed differently, as it is a form of art that is associated with greater cultural elitism than the museums of fine arts, history or popular art. The restriction of access to the public to this type of exhibition demands important mediation that mostly leads to the reduction in communities of artists and the community of researchers and specialist spaces. While institutions such as the National Center of Contemporary Art located in the commune of Cerrillos—far from the great cultural centers and with significant levels of poverty and vulnerability—do carry out some actions to build a connection to the territories, it is mostly oriented towards the artists and researchers of this field who are assumed as its community. For its part, the Museum of Contemporary Art is located in a central area of the city next

to the National Museum of Fine Arts, which is an institution with a wider-ranging public due to its historic importance. It promotes a diversity of educational and mediational actions, focusing on central themes of contemporary local art with a special emphasis on the relationship between art and new technologies. The workers of this museum state that they perceived a greater interaction with the public during the pandemic, raising the expectation of the widening of the territorial and community scope of this institution in the future.

From the aforementioned, we can see that, in general, it is the workers who, in their daily institutional practices, carry out the tasks of giving form and voice to the imagined community of museums, especially national ones, while seeking to overcome existing gaps and generating greater proximity of the institution to different groups and members of the public, i.e., artists, researchers, neighborhood communities, children, students, elderly people, etc.

More specifically, the aspects highlighted by each institution on this issue can be summarized in the following manner, as we can see in Table 3:

**Table 3.** Museums and Communities.

| Institution | Central Issue |
|---|---|
| National Museum of Fine Arts | Development of an inclusive and territorially linked policy through the deconstruction of the 'nineteenth century' aspect of the museum, from curation to collections, and primarily, educational efforts. |
| National History Museum | Change in the script of the permanent exhibition to redefine the underlying notion of 'national' in the museum, impacting the recognition of diverse and territorially expanded communities that it should address. |
| Museum of Contemporary Art | Search for virtual engagement in the wake of the pandemic, expanding the museum's audiences. Recognition of a broader audience consuming digital content, in addition to artists, researchers and students (a more permanent community of the museum). |
| Tomás Lago American Museum of Popular Art | Need to connect with the artisan community. Generation of institutional actions to expand connections. Seeking social relevance by embracing political and social actions of the upheaval, broadening the notion of the popular, and consequently, its community. |
| National Center for Contemporary Art | Reduced and more specialized audience. |

## 6. Conclusions

During the writing of this article, we sought to research global debates on a new definition so that museums resonate with daily life and the practices of public museums in Chile. To that end, from the perspective of institutional analysis (IA), we availed the reflections of museum professionals on their perceptions, necessities and immediacies regarding the present and the future of museums.

Among the relevant findings, we identified a series of issues of interest for the museums' teams: the need for a permanent financing policy; the absence of professional and academic training for the sector; the scarcity of articulation between the various units of the museum; the need to foster greater association and generation of networks; the demand for virtualization; the importance of experience compared with international cases; the need for greater inclusion of minorities (immigrants, indigenous people, LGBTQ+, children, etc.) and the need to implement strategies for forging links with territories and communities.

The findings presented in this paper were the results of theoretical–practical work carried out with the professionals of the main museums of the country who exchanged positions regarding how broader social processes have become of increasing importance and have caused transformations in the institutions. Likewise, the debates revealed

how changes in professional practices are negotiated and discussed between members of the institution.

The international debate on the redefinition of museums is articulated through the day-to-day practices of the workers, which imply issues particularly relevant to their working lives such as job insecurity or the lack of resources to carry out their activities, as well as the more abstract debates regarding the relationships between the museum and the community or virtualization. This reveals different levels of reflection and a constant transition between the theoretical–epistemological reflection on professional practices and the day-to-day decisions demanded by their institutional duties.

The research carried out caused us to distinguish new intelligibilities regarding how the social, political, economic and environmental transformation of the last few decades might be establishing a new relationship between museums and society. In future articles, we hope to go into greater depth into issues that were not raised in the present work such as decolonization (restitution of pieces, rewriting of scripts, inclusion of indigenous peoples, etc.) or the incorporation of feminist and intersectional perspectives in the museum.

Likewise, on a local level, it is in the interest of all concerned to continue this line of research to comprehend the transformations in Chilean museums following the social uprising of 2019 and the failed constitutional processes that have left the demand for greater citizen participation in limbo.

Finally, on both a global and local level, two issues to be urgently analyzed have arisen regarding the museum: the climate crisis and the rise of new forms of authoritarianism which raise the question of the place of institutions of culture and heritage in order to value democracy and both cultural and natural diversity. Phenomena such as this cause us to continue the reflection that began here on the social roles of the museum as an institution with the great potential to imagine and anticipate a future of greater solidarity and sustainability.

**Author Contributions:** Conceptualization, M.F.M. and R.R.C.; methodology, R.R.C.; formal analysis, M.F.M.; investigation, M.F.M. and R.R.C.; data curation, R.R.C.; writing—original draft preparation, M.F.M. and R.R.C.; writing—review and editing, M.F.M. and R.R.C. All authors have read and agreed to the published version of the manuscript.

**Funding:** This research was funded by the University of Chile through the UCH-1899 project.

**Institutional Review Board Statement:** Not applicable.

**Informed Consent Statement:** Not applicable.

**Data Availability Statement:** The datasets presented in this article are not readily available because the data are part of an ongoing study. Requests to access the datasets should be directed to the authors, who will be able to provide more information once the study is completed.

**Conflicts of Interest:** The authors declare no conflict of interest.

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
