# Peer review of "The Museum as a Laboratory: An Approach to the Experience of Public Museums in Chile"

_socsci, doi:10.3390/socsci13020090_

Round 1

Reviewer 1 Report

Comments and Suggestions for Authors

The paper presents a well-established and pertinent contribution to museology reflection, supported in real cases of study. The authors present a topic of relevance and contextualise their research in academic literature, creating an interesting article. The main concerns about the paper are regarding the assimilation of the paper to the results of the "LAB_Museums: Contemporary Museums and Museologies" project, which at different points remains unclear. I recommend to clearly state if the paper is a publication of the results of this project or if it is a primary source, upon which they have constructed a further investigation. This question should be perfectly defined in the introduction, method, results, and conclusions. Furthermore, additional information concerning the mentioned project should be provided (number of institutions participants in the study -apart from the 5 museum institutions, other academic or governmental? -, coordination, scope, duration, and dates, etc.). Finally, results could be presented and synthesised more clearly, it engages the authors to create at least one summarising table with them, as well as differentiating the partial results (institution by institution) from the general ones. Some clues could be advanced by the authors to show how could this investigation and method be extrapolated to other museum networks. If these considerations are reviewed, I presume that the future readers would be able to ponder the precise contribution of this study. I acknowledge the overall merit of this work.

Author Response

The comments made by the reviewer are of great importance for the improvement of the text, especially regarding the presentation of data. We agree with the reviewer that we lacked better specification that the results presented are not final but preliminary information from an ongoing research. We welcome all the suggested changes, which we consider highly relevant, and have included the necessary modifications to address them.

Reviewer 2 Report

Comments and Suggestions for Authors

see attached file

Comments on the Quality of English Language

The English is in general very good and clear.

Author Response

The reviewer conducted an extensive evaluation of the text, from which we will embrace some suggestions and engage in a debate/dialogue with the reader of our text on others. Among the points on which we agree with the evaluation, we highlight the need to make the discussion we want to develop clearer, emphasizing the specific issues that affect Chilean museums and the local responses that professionals in these institutions develop in dialogue with international debates. Perhaps there was a lack of clarity in presenting the data and information, leading the reviewer to understand that the focus of our discussion would be to compare Chilean museums with those of the global North, demonstrating that they are "different" in institutional terms. On the contrary, our purpose is to observe, based on professional practices, how tensions between different discourses, of local and international origins, influence their work routines and, more broadly, impact the institution, leading to processes of transformation (or not). The reviewer mentions this objective at some point but does not recognize it as the main goal of our text. Thus, we will incorporate the literature suggested by the reviewer, especially on Institutional Analysis, our theoretical focus, to make our pursued objective clearer and more demonstrable to future readers. On the other hand, the criticism directed at the sample used in this article does not take into account that this was only a first stage of the project (a point we clarified in the revision process) and that there is academic interest in assessing how national museums, considered conservative and elitist, are being affected by changes in the museal field paradigm, as well as by socio-political issues in their territorial surroundings. The significant processes of institutional self-reflection - to become more egalitarian and democratic - that these institutions have been undertaking in recent years are of great importance for the analysis of both how institutional changes occur and their social consequences/effects. Finally, as commented by the reviewer, this text was written the way we elaborated it and not as he would like it to be written. However, we appreciate and have included all comments that we consider important to improve our version.

Reviewer 3 Report

Comments and Suggestions for Authors

This article looks at whether recent debates on social issues and museums coming from the "West" are relevant to museums in Santiago, Chile. The authors use Institutional Analysis and focus groups with museum professionals at 5 institutions to draw out a series of reflections on issues/concerns that museums face in Chile. As someone who used to work in a museum, and is still in touch with many museum professionals, I found  the quotes from museum professionals here expressed concerns that are very familiar and longstanding. While they have answered the research question they set for themselves, I do not feel that the authors were ambitious enough in their aims with this paper. Their results touch on a more or less universal set of institutional concerns faced by institutions around the world. The result they discover, that Chile's institutions face the same issues as institutions elsewhere, merely repeats a lot of existing studies cited here. The fact that institutions have debates on the use of the internet to reach a broader audience is not an original finding. The section on Communities (why is this heading highlighted?) is more promising. For a national museum to consider the communities it serves is an important development but the authors did not consider the idea that, among their sample of museums, there might be very different communities at stake (a popular art museum versus a contemporary art center versus a national history museum). In fact there were only quotes from one institution in this section. The scope of quotations and the discussion of them could be expanded considerably.

However, in the conclusion, the authors bring up excellent, pressing contemporary issues, like decolonizing museums, the climate crisis, and the rise of new forms of authoritarianism that could all be productively explored but they seem to think that these issues cannot be the subject of the current article. I suggest the authors make more of the distinctions among their sample museums and reconsider their frame, to push on the social/political issues further in order to develop a more robust argument that will offer a unique contribution to the literature as opposed to a well-written and generally well-cited piece that does not really open new territory. 

There are also some errors. I found like one short, untranslated paragraph (lines 344-348) and at least one citation that was not included in the References (line 73, Fannon--also misspelled).

Comments on the Quality of English Language

The quality of the English is generally strong, aside from the untranslated lines cited above.

Author Response

The comments from reviewer three were extremely helpful in revising our priorities in the article and making the social debate of the text more evident. We welcome the suggestion to reassess our analytical framework and have modified the text to place more emphasis on the data demonstrating how professionals in Chilean museums faced local political and social contingency processes and how this context led to internal debates within the institutions about their own institutional structure and social relevance. Additionally, as suggested, we have made the sample of museums worked on more explicit in the text to clearly highlight the conflicts/debates within the institutional context of each institution.

Round 2

Reviewer 2 Report

Comments and Suggestions for Authors

Situating this article in the context of the literature of IA rather than of primarily of theoretical museology makes its value and originality much clearer. The detailed explanation of the methodology, the history of the origins of the project and the more explicit assessment of the focus groups in terms of the theory of IA have all deepened its analytical approach, sharpening its relevance and insights.

Comments on the Quality of English Language

Just a final check by a native English speaker

Reviewer 3 Report

Comments and Suggestions for Authors

These authors have done an exceptional job in their revisions. It is a much more effective and far-reaching presentation that makes important points about the reconsideration of museum presentations and processes in Santiago as a result of the 2019 protests and the onset of Covid. Really nice work and congratulations to the authors. Well done!

Line 414 has a typo.